# Prevention of Cardiovascular Diseases in Community Settings and Primary Health Care: A Pre-Implementation Contextual Analysis Using the Consolidated Framework for Implementation Research

**DOI:** 10.3390/ijerph19148467

**Published:** 2022-07-11

**Authors:** Naomi Aerts, Sibyl Anthierens, Peter Van Bogaert, Lieve Peremans, Hilde Bastiaens

**Affiliations:** 1Department of Family Medicine and Population Health, Faculty of Medicine and Health Sciences, University of Antwerp, Wilrijk, 2610 Antwerp, Belgium; sibyl.anthierens@uantwerp.be (S.A.); lieve.peremans@uantwerp.be (L.P.); hilde.bastiaens@uantwerp.be (H.B.); 2Centre of Research and Innovations in Care, Department of Nursing and Midwifery, Faculty of Medicine and Health Sciences, University of Antwerp, Wilrijk, 2610 Antwerp, Belgium; peter.vanbogaert@uantwerp.be

**Keywords:** implementation science, contextual analysis, qualitative research, primary health care, community, interview, focus group, cardiovascular diseases, prevention, Consolidated Framework for Implementation Research

## Abstract

Cardiovascular diseases are the world’s leading cause of mortality, with a high burden especially among vulnerable populations. Interventions for primary prevention need to be further implemented in community and primary health care settings. Context is critically important to understand potential implementation determinants. Therefore, we explored stakeholders’ views on the evidence-based SPICES program (EBSP); a multicomponent intervention for the primary prevention of cardiovascular disease, to inform its implementation. In this qualitative study, we conducted interviews and focus groups with 24 key stakeholders, 10 general practitioners, 9 practice nurses, and 13 lay community partners. We used adaptive framework analysis. The Consolidated Framework for Implementation Research guided our data collection, analysis, and reporting. The EBSP was valued as an opportunity to improve risk awareness and health behavior, especially in vulnerable populations. Its relative advantage, evidence-based design, adaptability to the needs and resources of target communities, and the alignment with policy evolutions and local mission and vision, were seen as important facilitators for its implementation. Concerns remain around legal and structural characteristics and intervention complexity. Our results highlight context dimensions that need to be considered and tailored to primary care and community needs and capacities when planning EBSP implementation in real life settings.

## 1. Introduction

Cardiovascular diseases (CVD) are the number one cause of death; more people die annually from CVD than from any other cause. In 2019, an estimated 17.9 million people died from CVD, representing 32% of global mortality [1,2]. According to estimates of the WHO, nearly 75% of premature CVD deaths are preventable [3]. The current literature demonstrates numerous methods to reduce CVD risk with strong consensus on the importance of raising awareness of CVD risk factors and the asymptomatic course of CVD, and on the impact of health behavior and lifestyle on health outcomes [4,5]. The burden of CVD is highest among individuals in the lower socioeconomic status (SES) quintile as a strong relationship exists between cardiovascular health and education level, occupation, and income [6,7]. Studies suggest that where improvements in CVD-related outcome have occurred, there is an inequity in benefits with a lesser impact on those people of lower socioeconomic status [8]. In order to increase quality and accessibility of care [9,10,11,12], new models of primary health care (PHC) are needed [13,14] and primary prevention should be an important priority for health policy makers [15].

Health systems are reorienting towards health promotion, defined as “the process of enabling people to increase control over, and to improve their health” [16], and disease prevention. Nurses play a critical role in expanding, connecting, and coordinating primary and community care [17] and have the ability to make a difference in areas such as patient advocacy and education, and people-centered care [18]. Clinical practice guidelines recommend active and systematic integration of lifestyle interventions for CVD prevention in PHC and community settings [19], adding to the importance of integrated care by general practitioners (GP) and practice nurses (PN) in general practice [20,21]. Such collaborations are only established to a limited extent in some contexts [22]. The benefits of a nurse-coordinated approach on morbidity, mortality, and lifestyle-related risk factors in both primary [23,24,25] and secondary [26,27,28,29] prevention of CVD have been demonstrated. Moreover, evidence also shows intervention models that have successfully used peers and community partners as facilitators to enhance health [30,31], and that they can be trained for CVD prevention and management in a cost-effective manner [32].

In Flanders, Belgium, only an estimated 30% of general practices are supported by a PN, and the job profile and ethical framework remain insufficiently clear [22]. Furthermore, the link between PHC and the community is unclear. A reform of the health system is ongoing to establish the basis for strong integrated care and strengthen well-being initiatives, social care and health care and their interaction [33].

Studies show poor achievement of guideline-recommended CVD prevention targets [34,35], as the translation of evidence-based interventions to practice remains limited. Moreover, little is known about how context (Definition: context reflects a set of characteristics and circumstances that consist of active and unique factors that interact, influence, modify, and facilitate or constrain the intervention and its implementation) can influence their implementation [36,37]. As such, there is an urgent need to investigate the factors that could facilitate or hinder the implementation process in specific primary care and community settings; and with our study, we provide an approach transferable to other contexts. These insights will allow us to further contextualize and plan the implementation process of targeted interventions and strategies for detection and management of CVD risk factors in the general population as well as in vulnerable subpopulations. In addition, we provide recommendations for planning successful and sustainable implementation.

The primary aim of this study was to explore macro-, meso-, and microlevel stakeholders’ views on implementation determinants of a comprehensive intervention for the primary prevention of CVD prior to its implementation in general practice and community settings. A secondary aim was to foster buy-in and sustainability through stakeholder engagement.

## 2. Materials and Methods

### 2.1. Study Context

This pre-implementation contextual analysis is part of the H2020 SPICES project, which intends to scale up packages of interventions for cardiovascular disease prevention in selected sites in Europe (France, UK, Belgium) and Sub-Saharan Africa (Uganda, South Africa). The Consortium developed the significant components of the intervention, referred to as “the evidence-based SPICES program (EBSP)”, based on systematic reviews of international guidelines [19,38]. The first component is risk profiling and communication for people between 40 and 75 years, applying the non-laboratory INTERHEART modifiable risk score [5]. The second component comprises multi-lifestyle-behavior change counseling (BCC) for those at medium risk with follow-up for at least one year, focusing on the DASH diet, combined aerobic training or aerobic and resistance physical activity, and smoking cessation. Finally, the Consortium decided to incorporate at least the following behavior change techniques in the interventions: motivational interviewing, goal setting, action planning, and problem solving.

### 2.2. Study Design

We conducted this qualitative study within a transformative research paradigm which provided the participatory philosophical assumptions behind the change-oriented SPICES project [39,40]. The EBSP served as the basis to go into dialogue with our local key stakeholders to ensure its components and target implementation strategies [41,42,43] to take form through co-creation. Inspired by the WHO’s Innovative Care for Chronic Conditions Framework, we focused on the partnership triad consisting of patient and family, community partners (CP), and PHC team [44,45]. We also selected the Consolidated Framework for Implementation Research (CFIR), a determinant framework, to guide our data collection and analysis and the reporting of our results [46,47,48].

### 2.3. Study Setting and Purposeful Sampling

This is a two-phased study. In the first phase, we performed key stakeholder identification and mapping during brainstorming sessions with the input of our local advisory board. In addition, we used the snowballing technique to identify additional key informants [49]. We included key stakeholders on the national (Belgium), regional (Flanders), and local (city of Antwerp) level where the EBSP was planned to be implemented. Key stakeholders included relevant central and local government organizations and agencies, policy makers, nongovernmental and community-based agencies involved in the implementation of CVD services, development partners and study communities, representatives of patient organizations, PHC providers, and coordinators of welfare organizations.

In the second phase, we purposefully selected a heterogeneous sample of the organizations at PHC and community level, eligible for future implementation of the EBSP in Antwerp. This process was carried out in consultation with key stakeholders from phase one who were familiar with the study context. We also used snowballing strategies; consulted the platforms of professional networks or associations; and utilized pre-existing networking structures. Local organizations were found eligible if they could facilitate reaching vulnerable populations (i.e., low SES). We only listed community health centers (in Belgium, a ‘community health center’ is a multidisciplinary PHC team which is embedded in a third-payer financial system, thus making PHC accessible for vulnerable populations) or general practices if they confirmed a planned or existing formal collaboration with a PN at the time of the study and if they were organized as a group practice. Welfare organizations needed to be nonprofit and have a clear social engagement.

We contacted the selected respondents and organizations by e-mail and telephone to inform them about the study. Contact persons were asked to identify one or more appropriate stakeholders within their setting to participate in this study.

### 2.4. Data Collection

In phase one, we held focus groups for primary data collection with the available respondents, to raise a discussion between the stakeholders from different fields of expertise. In addition, we conducted individual interviews with respondents who could not attend one of the focus groups and with the stakeholders we recruited during phase 2. We involved some of the stakeholders through informal meetings, of which we kept meeting reports. We developed flexible, semi-structured data collection tools to guide the interviews. In consensus with the international SPICES consortium, we developed the topic guides using the CFIR interview guide tool [50], which we further adapted to our local context and stakeholder groups. The interview guides are available in Appendix A. An experienced team of qualitative researchers (NA, SA, HB) collected data until we reached data sufficiency. At least two researchers were present as moderators or observers in each focus group. All interviews were held face-to-face and were audio recorded. The interviewers took field notes of their experiences during data collection.

### 2.5. Data Analysis

We applied a descriptive, adaptive framework analysis with a mixed inductive and deductive analytic approach [51,52]. Verbatim transcripts were read several times to familiarize with the data and to generate analytic memos and reflections. One researcher (NA) conducted an inductive, open coding on the transcripts of six individual interviews and two focus groups. Transcripts were divided into meaningful segments that were assigned with open codes, which were then grouped around various aspects regarding the research topic, resulting in clusters of interrelated subthemes and themes. The research team then further refined this inductive preliminary coding structure (NA, SA, LP, HB). In the next step, we charted our preliminary coding structure into the CFIR by mapping interrelationships with domains and constructs [53]. Operational definitions of CFIR domains and constructs were tailored to the study to improve coder consistency [Appendix B]. This iterative and reflective process required several discussion rounds within the research team (NA, SA, HB) and resulted in the adaptive analysis framework that we used to deductively code the remaining transcripts. Microsoft Excel 2016 software supported the charting of the data which involved summarizing the data by domain and construct or category from each transcript. The framework was flexible to new findings, thus it was regularly discussed and adapted when needed in team discussions [54] (NA, SA, HB). Finally, we triangulated the data from the study phases and sources by carrying out a framework-focused document analysis of the meeting reports to further substantiate our results [55]. 

The first study phase ran from July 2017 to December 2017 and the second study phase ran from November 2018 to April 2019. To meet the overall quality standards, we followed the COREQ checklist [56] for reporting the results of this study.

## 3. Results

In phase 1 of this study, 24 key stakeholders participated, as outlined in Table 1. In phase 2, lay CPs, GPs, and PNs from 4 welfare organizations and 12 general practices were involved. The characteristics of the included primary care settings and welfare organizations and their respondents are outlined in Table 2 and Table 3, respectively.

Our main findings are further reported below, according to CFIR domains and relevant constructs. A comprehensive summary of the results is provided in Figure 1.

### 3.1. Intervention Characteristics

#### 3.1.1. Relative Advantage

Our respondents indicated that implementing the EBSP will result in increased detection of people at risk that are currently missed for prevention. The combined strategy of implementing the EBSP in both general practices and in nonclinical community settings is expected to improve reaching vulnerable people for prevention.


*“I think we reach many people with certain risk factors. So that is an advantage, because otherwise they are isolated… it concerns people who do not take the steps towards health care, who don’t find their way there.”*

*[CP]*


The EBSP could give general practices the opportunity to improve current preventive practice by its systematic and structured implementation. Involving CPs, as well, is expected to reinforce integrated care with a holistic approach and will demedicalize CVD prevention. The opportunity to link the currently fragmented initiatives in PHC and community settings is considered a strong advantage of the intervention.

#### 3.1.2. Adaptability and Trialability

Re-evaluating and adapting the EBSP to each setting’s specific characteristics is seen as critical throughout each phase of the approach, so that it can be embedded in current workflows and systems. Potential implementers need the possibility to test the EBSP on a small scale, allowing them to iteratively co-create, test, and modify the intervention components and implementation strategies to their needs and preferences.


*“In some settings it will run smoothly, but in other settings it just won’t. We will then have to see how that fits into our system here. You have to start somewhere, of course… and then maybe re-evaluate and adjust it if necessary.”*

*[CP]*


#### 3.1.3. Complexity

EBSP components vary in complexity; risk profiling and communication were estimated to be low in complexity, whereas BCC was predicted to be very complex, especially in vulnerable populations. Our respondents believed that the tools developed to support the EBSP are user friendly. Especially the selected profiling tool was considered easy to incorporate since it is clear and does not include sensitive questions. However, the measurement of hip-waist circumference is not common practice in community settings and could pose a barrier due to role confusion.


*“I wonder whether people who come to a community center would appreciate having their waist circumference measured there by a social worker.”*

*[GP]*



*“Behavior change is a very difficult thing. In my experience, I find that people rarely do really change their behavior…”*

*[PN]*


Furthermore, our respondents felt that medical lay people do not have the appropriate profile to perform the complex BCC component, as this requires specific competences that cannot sufficiently be developed through a project-related training package. An extended PN role was believed to fit with all EBSP components in the general practice.


*“Profiling is not carried out systematically in the general practice, not even for those health-related topics where it is perfectly feasible. And in our context, we don’t have the volunteers at community level... so who’s responsibility will it be?”*

*[Team leader dept. prevention, Flemish Government]*



*“The role of a ‘PN’ doesn’t exist in every general practice yet, and each practice autonomously decides how that PN will be deployed exactly.”*

*[Team leader dept. prevention, Flemish Government]*


### 3.2. Outer Setting

#### 3.2.1. Population Needs and Resources

Our respondents suggested to clearly define vulnerability for CVD based on the presence of CVD risk factors, lack of awareness of individual risk, and SES. They recognized the link between a low SES and poor health status, unhealthy lifestyle and habits, very limited access to health care, and low health literacy.


*“The majority of people at high risk is not aware of it, because often these risk factors give little or no complaints and the GP is not systematically consulted to have this checked.”*

*[Managing director National cardiologists association]*



*“People who live in poverty or who do not speak the language are less able to pick up information.”*

*[PN]*


Meeting the needs of the target population was an important implementation driver for potential settings and implementers of the EBSP. Respondents stressed the need to empower the target population to take informed health decisions by raising awareness for the prevention of CVD. However, they also discussed the financial, practical, and cultural challenges of reaching a vulnerable population for prevention. Respondents expressed their concerns around the relative priority of prevention in relation to multiple dimensions of the complex context around vulnerable populations.


*“They disappear under the radar, and then reappear when they have an acute problem, where you don’t really have the time for education.”*

*[GP]*



*“Someone who does not have proper housing, does not have the mental capacity to discuss health.”*

*[CP]*


The EBSP should take a broader approach of health promotion, rather than focusing solely on CVD prevention. Patient advocacy is needed, especially in vulnerable populations requiring extra guidance and navigation to quality health care. Respondents also raised the need to support and empower people to become active participants of their health, e.g., by improving health literacy and self-management support.


*“Poverty is mainly about social exclusion. And that’s why, when you want to activate people towards regular care, it needs much more effort from us to get those people there and to keep them there.”*

*[Coordinator Association for people in poverty]*


#### 3.2.2. Cosmopolitanism

Our context was described as a fragmented landscape of preventive care, with parallel initiatives at PHC and community level. The level of collaboration with external partners strongly varies amongst organizations and although certain forms of collaboration exist, formal collaborative structures are currently lacking.


*“A whole network is formed around certain populations, with many actors all acting in related domains… in parallel, often without knowing about each other.”*

*[Coordinator Association for people in poverty]*


Respondents stressed that a shift towards network-oriented care is needed, urging better alignment of mission, vision, and goals. They recommended to primarily implement the EBSP in regions where the basic conditions for such a network are already fulfilled and to strengthen and scale up the link between existing initiatives and actors to enhance the impact on larger communities.


*“If people are not working together in a good way, it will be difficult to launch a project like this. You should focus on regions where there is already a good collaborative network between different actors, based on mutual trust and know-how.”*

*[Pharmaceutical Care Coordinator]*


#### 3.2.3. External Policies and Structures

Our respondents highlighted the compatibility with the ongoing macrolevel reform of PHC, with policy makers supporting the transition towards integrated care, prioritizing interdisciplinary collaboration within a person-centered care model. However, the extent to which the EBSP can be implemented in community and PHC settings, depends on the resource capacity of organizations, local policies, and national guidelines. Organizations might be restricted when participating in the EBSP given the lack of clearly defined complementary responsibilities in preventive care and related financial compensation. 


*“The political government must continue to provide budget for us to be able to continue our preventive care initiatives… Unfortunately, the priorities are not always the same.”*

*[Team leader dept. prevention, Flemish Government]*



*“The Flemish GPs Association has developed a very nice prevention plan, however, it doesn’t seem to get implemented in practice. There is just no time and it is not reimbursed.”*

*[GP]*


### 3.3. Inner Setting

#### 3.3.1. Implementation Climate

A need for change arises from dissatisfaction with the current approach to preventive care, which does not allow to adequately respond to changing care demands. It was emphasized that a holistic view of social and other determinants is needed to improve the overall well-being of people through strong partnership between welfare and PHC. Close collaboration and clear definition of complementary responsibilities and job contents through protocol care to guide interdisciplinary partnerships including task delegation and task shifting, were mentioned as facilitating factors. However, complex collaboration implicates difficulties in the organization of work processes, communication, keeping vision and mission aligned, and decision making, all of which could impact the EBSP implementation.


*“It is often the case that the future situation of a person is disease-related, thus health is or will always be an issue for us as well. This could be a motivation for organizations like ours to participate in this project.”*

*[CP]*



*“When it comes to shared responsibility, protocol care is so important.”*

*[PN education coordinator]*


The compatibility of the EBSP was reflected in its fit with norms, values, needs, existing workflows, and systems of eligible partner organizations. Therefore, the vision of partner organizations should contain aspects from the EBSP, such as focus on prevention; interdisciplinary collaboration and task delegation; accessibility and inclusivity of care; and outreaching community activity. In that case, it would be feasible that existing workflows are redesigned with the EBSP.


*“We collaborate with our PN, who take the time to take up preventive tasks. In other practices, less time is invested in prevention. Care providers must also be open to work with a vulnerable population, and I am afraid that this is not always the case.”*

*[GP]*



*“It could also turn out to be a great advantage that in our practice nothing has really been developed structurally around prevention, and that with this project we would be given the opportunity to translate our plans into something actionable… and also for me to expand my role as a PN.”*

*[PN]*


Respondents expressed their concerns around existing higher priority responsibilities posing a potential threat to the EBSP in both PHC settings and community settings. 


*“PHC is overburdened, we really feel this at practice level. Because of a high workload, prevention is often the first thing that is neglected.”*

*[GP]*


#### 3.3.2. Readiness for Implementation

Next to active involvement and engagement from formal and informal leaders, the EBSP will need to be supported by the whole team involved in its implementation.


*“According to our team leader, you cannot expect that the EBSP will be implemented, because the necessary time commitment cannot possibly be guaranteed by the managers.”*

*[CP]*



*“It is also important for everyone to be open to new things, because one person who does not feel up to it can jeopardize the whole project.”*

*[PN]*


Our respondents anticipated some challenges around availability of resources in potential implementation settings. A high workload and the lack of structural financing for the cost of the implementers’ dedicated time could hinder the implementation. Introducing creative solutions to facilitate interdisciplinary collaboration will be needed to increase the capacity to systematically implement the EBSP: e.g., task delegation and supportive financial systems and incentives. 


*“We chose to work under the capitation payment system from the beginning, which means that we are able to delegate a number of tasks to the PN who we supervise. But I must say that prevention is being put aside because there is simply no time for it at the moment.”*

*[GP]*


### 3.4. Characteristics of Individuals

#### 3.4.1. Knowledge and Beliefs about the Intervention

Our respondents indicated that it will be important for all actors involved to have confidence in the EBSP. They expressed a positive attitude, but some were skeptical towards obtaining actual behavior change as a health outcome, especially in vulnerable populations.


*“Behavioral change is very difficult...In my experience, people rarely really change their behavior. Motivation is something that has to come from the people themselves.”*

*[GP]*


#### 3.4.2. Self-Efficacy

The diverse backgrounds of potential implementers will determine their level of pre-existing competences in EBSP components. Respondents showed confidence in the competences required to perform the risk profiling using the project tools provided. However, they lacked confidence in the knowledge and skills related to risk communication and BCC techniques and stressed the major need for specific training in all EBSP components.


*“During my studies, subjects were discussed about counseling groups and individuals... but most of the actual know-how you get from practice, I think.”*

*[CP]*



*“I think we should organize more training within the practice. That is actually a permanent need.”*

*[GP]*


### 3.5. Implementation Process

#### 3.5.1. Planning

Respondents recommended developing a structured action plan together with the potential implementers. In addition, implementer interrelationships, including communication, knowledge sharing, team-oriented problem-solving, and structuring collaborations through care plans, will be needed to accomplish successful implementation.


*“In order to get something running in the practice, you have to sit together regularly with systematic follow up. That’s also crucial for thorough planning and structurally incorporating the EBSP.”*

*[GP]*


#### 3.5.2. Engaging Implementers and Intervention Participants

Long-term and sustainable partnerships will be challenging to develop and maintain. Respondents advised to use bottom-up and participative, collaboration-oriented strategies, alongside creating local project visibility, participating in structural platforms, investing time and effort to engage local organizations, and staying connected with implementers during each phase of the process.


*“A participative approach, being in it, and creating it together -certainly not top-down… but growing something bottom-up.”*

*[Team leader dept. prevention, Flemish Government]*



*“We should find ways to see that anything you will achieve with SPICES gets anchored, instead of losing everything that you built in the field.”*

*[Health promotion coordinator, Primary care network]*


Our respondents proposed to select and combine various recruitment strategies together with micro- and mesolevel stakeholders to overcome barriers in reaching vulnerable populations for preventive initiatives. Most importantly, interventions should be implemented in a familiar and psychosocially safe environment through the established trust-based relationship with the target population.


*“We see that the role of the GP is crucial for our people. The GP is also a person they trust. It is the one person from the medical world they have the most confidence in, and who they can really talk to.”*

*[Coordinator Association for people in poverty]*


A combination use of active and passive communication channels was suggested. Activation of the social network around people, and intensive and personal referral and navigation of people towards community initiatives or health care, will facilitate the reach of participants.


*“There is always someone from our organization that goes with them the first time. This way, the familiar and trusted environment comes along wíth them really. And we also try to make sure that they receive a warm welcome on the other side as well… You know, our people are so suspicious of everything that is unknown.”*

*[Coordinator Association for people in poverty]*


Working together in a participatory way with vulnerable people requires a sincere and open attitude towards their context. A barrier to the intervention could be that health care providers often lack the time to provide the follow-up that is needed to keep them involved long-term.


*“A participatory approach is crucial. If you take people seriously, from the outset, about their story and what they encountered and what they think could be solutions, that’s a very important first step.”*

*[Coordinator Association for people in poverty]*


Our respondents recommended the use of several communication and BCC techniques: such as, motivation to change, goal setting, result-oriented approach, shared decision making, tailoring messages, and supportive materials.


*“By emphasizing what’s in it for them, and if you start from the patient’s perspective, you will get much further.”*

*[GP]*


## 4. Discussion

This study explored the views of macro-, meso-, and microlevel stakeholders on the contextualization of a comprehensive intervention program for the primary prevention of CVD, along with determinants to its implementation in PHC and community settings in a Belgian urban context. This pre-implementation study was carried out as part of the H2020 SPICES project since contextual factors may be necessary for implementing the EBSP. The CFIR identified determinants, barriers, and facilitators across its domains and constructs, providing an opportunity to inform further design of intervention components and implementation strategies for implementation in new settings in the project’s next steps.

The SPICES project specifically intends to improve reaching vulnerable low SES groups for CVD prevention. Reaching people with low SES by health promotion and prevention initiatives on a population level is challenging [57,58]. Our respondents stated that a combined approach of implementing a CVD prevention program in both PHC and community settings is needed to increase accessibility to the EBSP and to affect the prevalence of CVD, which is further supported by the literature [19]. On the one hand, according to the literature, general practice plays an important role in reducing socioeconomic inequalities by maintaining a trust-based relationship, facilitating patient-centered communication and premising personal targets tailored to the local community context [59,60]. However, with regards to CVD, we also know that although detection levels of CVD risk factors by GPs may be improving, many people with increased risk remain undetected. PHC teams should therefore continue to use low-cost, practical approaches to detect people at risk [61]. On the other hand, previous research also demonstrates that relatively high levels of community engagement can be attained by introducing community-based CVD prevention programs [62], and that it has the potential to effectively reach under-served groups [63]. Community-based strategies previously have successfully led to an improvement in CVD risk factors [64], with especially positive impact on improving population knowledge on CVD and risk factors, physical activity levels, and dietary patterns [65,66].

Consequently, the SPICES project may offer the opportunity to link the currently fragmented landscape of PHC and community organizations by proposing CVD prevention as a common goal. Stakeholders indicated that coordination and proactive alignment between different policymakers and other stakeholders and adequate funding are fundamental for reorientation towards community-oriented care, which is in line with previous study findings [67,68]. Such a reform requires advocating for a mission and vision focused on integrated care, fostering collaboration with a focus on population care, regional multisector collaborative partnerships, and comprehensive strategies to transform health and well-being in communities [69,70]. The literature also suggests that community leadership, shared decision making, linkages with other organizations, and a positive organizational climate are key for building such partnerships [71].

The complexity of the SPICES project mainly lies in sharing responsibilities, especially when roles will be expanded through task shift and delegation to PNs in general practices and medical lay people in community settings. Optimizing the engagement of innovative providers requires clear definition of roles and scopes of practice, sensitization, in-service training, and formal supervision [72]. Trained nurses can easily take over preventive tasks without compromising quality of care and patient outcomes [73]. PHC can also be unburdened or supported by community approaches in implementing the EBSP. Previous research shows that community-based nurse-led interventions result in positive outcomes for patients with increased CVD risk. However, the success of such interventions needs to be facilitated by appropriate funding, thoughtful intervention design, and training opportunities for nurses [74]. Furthermore, in noncommunicable disease control programs, community health workers (CHW) deliver preventive services using informational as well as behavioral approaches worldwide [31,75]. However, this strong community component is not yet embedded in Belgium, implying such roles are currently not supported. Integration of such roles into the general healthcare system and existing community structures should be considered, taking into account population needs, health system requirements, and resource implications [76].

In addition, with the introduction of new roles, it will certainly be important to provide training in all components of the EBSP, especially with regard to BCC. Several studies show heterogeneity across the reporting of BCC training program content and structure, despite the importance of increasing providers’ competency to effectively counsel a population with increased CVD risk to change their lifestyle, and ultimately to improve healthcare services and health outcomes [77,78]. It will be important to properly explore the current competency levels and training needs of implementers, and to adapt the support from SPICES to fit. From previous studies, we do know that BCC training programs are mostly based on motivational interviewing and the 5 A’s approach, using multiple BCC techniques, and delivered through seminars and workshops presenting opportunities for interprofessional education [78,79]. Competences seem to be best acquired through active, realistic practice and implementation of reminder and feedback systems within actual clinical practice settings [80].

Adaptability will allow practitioners to improve current practice with evidence-based interventions which will be tailored, tested, and evaluated together with the implementers. Adaptability is indeed a crucial element in order to meet local needs, to address barriers and leverage facilitators, and to preserve fidelity [81,82,83]. It will be important to clarify the timing, context, and process of modifying interventions to facilitate their implementation, scale-up, and sustainment [84]. We will need to take into account the needs and specific characteristics of a vulnerable population and to adapt interventions and strategies accordingly. The literature shows limited lifestyle effectiveness of behavior change interventions for low SES populations [85]. Other studies highlight the urgency to tailor lifestyle interventions to the needs of vulnerable populations and call for health care providers and users to engage with behavior change techniques rather than focusing on information provision alone [86]. Effective interventions have a tendency to have fewer techniques [87].

Our respondents made some suggestions to take on in the next steps of the EBSP implementation, especially with regard to planning the implementation and engaging implementers and target population. We should take into account a thorough planning and implementer interrelations within the context of each organization. Previous studies show that organizational culture most commonly affects implementation and that leadership plays a crucial role in successful implementation of evidence-based practices [88]. Factors contributing to engaging and sustaining partnerships with microlevel implementers include starting small-scale and focused to build trust among participants, working within the framework of integrated preventive care, and providing long-term support [89]. Efforts to reach the vulnerable target group should be tailored and embedded in their familiar context, which is supported in previous studies suggesting face-to-face invitations from a reliable source and community outreach to raise awareness to facilitate participation [90].

We recognize that some limitations have to be considered when interpreting our findings. This study did not capture the perspectives of the target population; however, we did include stakeholders from organizations representing vulnerable groups. In the next steps of the SPICES project, it will be crucial to further explore members of the population’s perspectives. Furthermore, our sample might have been biased since we purposefully included stakeholders from organizations or settings with a link to the concepts of our project. On the other hand, including a large sample of stakeholders from different levels offered us the opportunity to critically triangulate our findings during the different study phases, increasing credibility. The methodology we used allowed us to give responsive feedback to the participants through member checking. It also reinforced the transferability of our results beyond this context by employing the CFIR as an established conceptual framework, further strengthened by the detailed description of the context of this study. The use of the CFIR ensured that all critical implementation determinants were explored, increasing the chance of successful and sustainable implementation of the EBSP. 

Our findings have the potential to inform the design and implementation planning of related health programs in similar contexts, and we have therefore translated them to key recommendations for planning successful and sustainable implementation as summarized in Box 1.

Box 1Recommendations for planning successful and sustainable implementation of a CVD prevention program.
➢Evaluate the unique context of a planned implementation and map potential barriers and facilitators. The CFIR is a useful tool to do so.➢Consider both general practices and welfare organizations as important avenues for primary prevention of CVD, especially when targeting vulnerable populations.➢Involve stakeholders, implementers and communities at all stages of the implementation, including project design and planning. Use participatory strategies to get and keep them engaged.➢Work towards stepwise implementation allowing adaptation to dynamic needs.➢Align intervention purposes with local policy, vision, and mission. Set achievable goals taking into account available resources.➢Design interventions in a way that they can be integrated in pre-existing workflows and systems.➢Offer support and develop tools mitigating the complexity of the intervention.➢Build networks between primary care and community partners.➢Explore collaboration models: practice nurses and lay community partners can play a critical role.➢Make sure that those who will provide the intervention have the necessary competencies or provide tailored training so they can be acquired.➢Generate ownership in members of local organizations.➢Take a broader approach of health promotion rather than focusing solely on CVD prevention.


## 5. Conclusions

Macro-, meso-, and microlevel stakeholders’ views demonstrated various contextual dimensions to consider when implementing a comprehensive program comprising complex interventions for the primary prevention of CVD in PHC and community settings and underscored several criteria that seem necessary to transform health systems towards a network-oriented approach of health and well-being. These results form a solid foundation to tailor the H2020 SPICES project to the needs and preferences of the target population and potential implementers, but also, to better respond to policy evolutions. The next steps in our research project can clarify how these complex and dynamic determinants are interrelated and how they influence the outcomes and process of implementing the EBSP in real life settings. Ongoing stakeholder engagement is needed to develop sustainability in this multidimensional, multilevel, and dynamic field.

## Figures and Tables

**Figure 1 ijerph-19-08467-f001:**
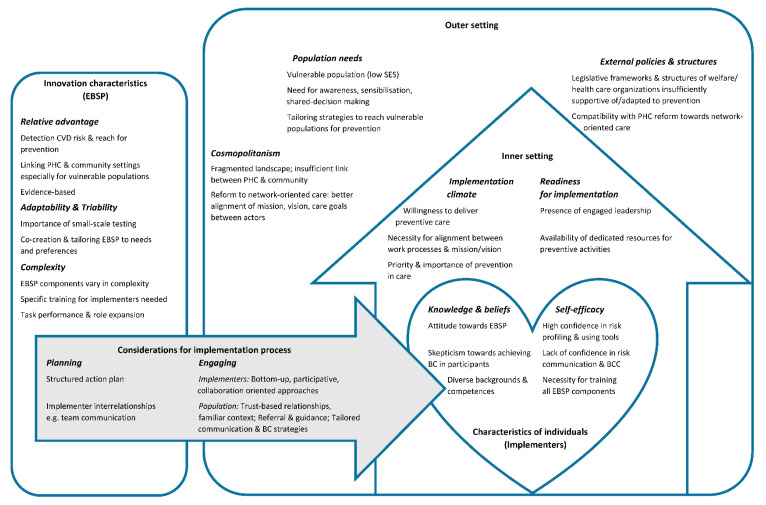
Summary of main results structured into CFIR domains and relevant constructs.

**Table 1 ijerph-19-08467-t001:** Phase 1 macro-, meso-, microlevel stakeholder characteristics (*n* = 24).

Stakeholder Level	Organization Type	Description Aims and Domain of Expertise	Job Description	Tenure in Current Organization (Years)	Data Source
**Macro Level**	Flemish Government-Dept. disease prevention (*n* = 3)	Department of disease prevention; related to health promotion and preventing diseases and disorders by (a) achieving the health objectives by implementing the accompanying action plans (e.g., healthy diet, physical activity, sedentary behavior), (b) recognizing and subsidizing partner organizations, organizations with field operations, loco-regional networks, (c) advising on and supervising a healthy environment.	Team leader Prevention Department	14	Interview
			Team member Prevention Department	10	Interview
			Head of Prevention Department	0.5	Interview
	City of Antwerp–Dept. health and welfare (*n* = 2)	Coordination of health projects with expertise in health inequity. Responsibilities regarding accessible health care: support and location of general practices (GP shortage and practice organization), promoting collaboration between welfare and health care partners, implementing health promotion and prevention, increasing access to care at community level and studying the use of the healthcare system.	Expert in accessible health care and health inequity	3	Focus group 2
			Healthcare Specialist: Health literacy and social health	1.5	Meeting report(s)
**Meso Level**	National cardiologists association	Information and exchange platform for CVD for patients. Primary and secondary prevention of CVD in the general population. Informing and early detection of CVD or risk factors.	Managing director	13	Focus group 1
	National health insurance organization	Expertise in health economics, public sector, data management. Coordination of research department. Innovation in health care networking and setting up projects.	Research and Innovation coordinator	20	Focus group 1
	Flemish general practitioners association	Promoting the interests of general practitioners in Flanders on a scientific, social, and syndical level through democratic decision-making and scientific foundation. Development and realization of a patient-oriented health care and policy. Expertise in prevention and health promotion.	Senior general practitioner coordinator	2.5	Focus group 1
	Primary care network	Networking organization, developing the Flemish government’s health promotion and disease prevention policy. Using evidence-based methods, offered by partner organizations, Flemish health objectives are translated in a sustainable manner into local and regional policy, actions, and projects.	Health promotion coordinator	3	Focus group 2
	Royal pharmacists association Antwerp	Professional association for pharmacists, developing the task of the pharmacist in health care and the pharmacist–population relationship. Supporting the patient in self-care and prevention.	Pharmaceutical Care Coordinator	3	Interview
	Local Multidisciplinary Network Antwerp	Local network supporting multidisciplinary cooperation. Improving quality of care for people with chronic disease: supporting caregivers, stimulating interprofessional collaboration, and increasing self-management competences of patients.	Care path promotor	1	Focus group 2
	Welfare linking organization in Antwerp	Focusing on exclusion due to poverty or origin by bringing people together. Providing opportunities for anyone experiencing exclusion. Experienced in reaching and working with people with low SES, setting up and running local projects on various (health) topics.	Senior regional volunteer	11	Focus Group 1
	General welfare center in Antwerp	Working on social challenges related to (dis) well-being. Central, innovative partner in welfare. Expert in working with vulnerable target groups. Aiming for equal opportunities in society.	Policy Coordinator Mental and Somatic Health, Migration	1	Focus Group 1
	Welfare and community development organization in Antwerp	Expert in working with socially vulnerable populations: people in poverty, social tenants, homeless people, single people, people without legal residence, low-skilled long-term unemployed. Fighting exclusion and disadvantage. Fundamental social rights as compass to realize structural changes: decent housing, education, social security, health, work, healthy environment, cultural and social development.	Team leader/coordinator	17	Interview
	Association for people in poverty	Networking organization. Negotiation between people in poverty, society, and policy. Bringing people in poverty together to work on structural changes that increase their quality of life. Bottom-up approach: meeting each other, sharing experiences, building networks, and starting actions and projects from their needs and preferences.	Coordinator	2	Interview
	Postgraduate training course ‘Nurse in the general practice’, University of Antwerp	Training course for nurses in specific general practice. Nurse autonomously supports GPs in treating, guiding, and caring for patients in primary care. Proactively responding to changing health care context.	Coordinator	2	Interview
	Flemish Institute for Healthy Living (*n* = 3)	Stimulating the population to live healthy in an accessible way. Providing practical advice, packages, and trainings. Partnering organization in prevention expertise of the Flemish government.	Staff member physical activity	2.5	Meeting report(s)
			Staff member general health promotion	1	Meeting report(s)
			Staff member general health promotion	0.5	Meeting report(s)
**Micro Level**	General practice A	PHC, working with vulnerable population.	General practitioner	1	Focus group 1
	General practice B	PHC, large proportion of patients are in the vulnerable group, working with prevention consultation in the practice.	General practitioner	8	Focus group 2
	Community health center A	Prevention (CVD amongst other diseases), culturally sensitive care, working with vulnerable groups (low SES).	General practitioner	5	Focus group 2
	Community health center B	PHC, working with vulnerable population.	General practitioner	2	Focus group 1
	Physical activity on prescription	Referral from GP to a certified physical activity coach. Helping vulnerable groups to live healthier and more active lives in an accessible way, starting from information from the GP and the needs and preferences of the participant.	Physical activity coach	0.5	Interview

**Table 2 ijerph-19-08467-t002:** Phase 2 primary health care setting, practice nurse, and general practitioner characteristics.

Primary Health Care Settings (*n* = 12)			Practice Nurses (*n* = 9)			General Practitioners (*n* = 10)		
Level of partnership between GPs	Community health center	3	Gender	Male	1	Gender	Male	4
	Duo practice	3		Female	8		Female	6
	Group practice	6	Tenure in practice (years)	>1	2	Tenure in practice (years)	1–2	3
Disciplines present, other than GP/PN	<3	5		1–2	5		>2–5	1
	≥3	7		>2–5	1		>10	2
Financial system	Fee-for-service	6		>10	1		>20	4
	Capitation payment	4	Postgraduate training	Postgraduate training	6	Data source	Interview	10
	Combination or other	2	Data source	Interview	9			
Level of PN involvement	Instrumental	5						
	Integrated	5						
	Planned in future	2						

**Table 3 ijerph-19-08467-t003:** Phase 2 welfare organization and lay community partner characteristics.

Welfare Organizations (*n* = 4)		
Organization Type	Description Aims and Domain of Expertise	Target Population
1. Community work	Focusing on social networking, community engagement, integration. Strengthening peer networks. Offering social and administrative support	Vulnerable adults: poverty, homeless, single, without legal residence, low-skilled unemployed
2. General welfare center community team	Focusing on welfare support (door-to-door, community centers). Working on social challenges related to (dis) well-being. Activities: crisis counseling, housing assistance, psychiatric care management	Highly vulnerable populations (SES, psychiatric, drug-related problems)
3. Social services	Public center for social welfare provides a wide range of social services and thus ensures the well-being of every citizen	People living in poverty, underprivileged children and youngsters, single parent families
4. Service center	Meeting place for local residents, offering information, recreation, training, and services. Outreaching welfare support in neighboring communities and service flats	Young seniors, (frail) elderly people and families
**Lay community partners (*n* = 13)**		
Gender	Male	3
	Female	10
Position in organization	Social worker	9
	Coordinator/team leader	4
Tenure in organization (years)	>2–5	1
	>10	2
	Unknown	10
Data source	Interview	3
	Focus group	10

## Data Availability

All data generated during this study are included in this published article. The original datasets analyzed during the current study are available from the corresponding author on reasonable request.

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
