# Peer review of "Prevention of Cardiovascular Diseases in Community Settings and Primary Health Care: A Pre-Implementation Contextual Analysis Using the Consolidated Framework for Implementation Research"

_ijerph, 2022, doi:10.3390/ijerph19148467_

Round 1

Reviewer 1 Report

Dear Authors,

Congratulations for this innovative and necessary work showing an evaluation of an implementation of an central preventive community program, additionally, in a field that requires complex and social strategies to change non-healthy behaviors, at the population level. One suggestion is to index in the Introduction, the concept of health promotion and the role of the Nurses at the community and primary health level, accordingly to WHO. Please, consider add this topics to be able to emphasize actors and actions in the future.

Good lock,

Reviewer 2 Report

Aerts et al., present a very well-written paper that could serve as a guideline for other healthcare professionals trying to set up similar interventional programs to combat CVD as well as other diseases that require the participation of multiple stakeholders and healthcare workers on a community level.

One addition that I believe would greatly help the readers is if the authors could add a brief section/table summarizing their findings and recommendations.

Reviewer 3 Report

1   1.      The introduction is very well grounded, but please emphasize the elements of originality/novelty that this study brings to the specialized literature a little more. I understand that the aim of this study was to explore macro, meso & micro level stakeholders’ views on the contextualization of a comprehensive intervention for the primary prevention of CVD prior to its implementation, so please:

·        Please insert the novelties that this study brings at the end of the introduction;

·        Please better highlight the purpose of this study.

2.      I want to mention that I have included the article to analyze the similarity coefficient in the Plagiarism CheckerX software, version 6.0.11, and please perform the following word reformulations that are in bold and italics:

·        This pre-implementation contextual analysis was carried out in the context of H2020 SPICES project, which intends to scale-up packages of interventions for cardiovascular disease prevention in selected sites in Europe (France, UK, Belgium) and Sub-Saharan Africa (Uganda, South-Africa).

·        Literature shows limited lifestyle effectiveness of behavior change interventions for low-income groups [81]. Previous studies also show that tailoring lifestyle interventions for this population is necessary, and more attention is needed to develop ways to ensure service providers and users engage with behavior change techniques, rather than focusing on information provision alone [82], and that effective interventions have a tendency to have fewer techniques [83].

3.      I have selected some essential paragraphs from the article for proofreading, and please change everything I suggest:

·        This pre-implementation contextual analysis was carried out (Passive voice) in the context of H2020 SPICES project, which intends to scale-up packages of interventions for cardiovascular disease prevention in selected sites in Europe (France, UK, Belgium) and Sub-Saharan Africa (Uganda, South-Africa). The major components of the intervention, referred to as ‘the evidence-based SPICES program (EBSP)’, were developed by the Consortium, based on systematic reviews of international guidelines [16, 33]. The Consortium developed the significant components of the intervention, referred to as ‘the evidence-based SPICES program (EBSP)’, based on systematic reviews of international guidelines [16, 33]. The first component is risk profiling and communication for people between 40-75 years, applying the non-laboratory INTERHEART modifiable risk score [5]. The second component comprises of multi-lifestyle-behavior change counseling counselling(BCC) for those at medium risk with follow-up for at least one year, focusing on DASH diet, combined aerobic training or aerobic and resistance physical physical resistance activity and smoking cessation. Finally, at least the following behavior  bahaviour change techniques needed to be incorporated (Passive voice) in the interventions: motivational interviewing, goal-setting, action-planning, and problem-solving.

·        In phase one, focus groups were held for primary data collection with the available respondents, to raise a discussion between the stakeholders from different fields of expertise. Individual interviews were conducted with additional respondents who were not able to could not attend one of the focus groups and with stakeholders who were recruited (Passive voice) during phase 2. Some of the stakeholders were involved (Passive voice) through informal meetings, of which we kept meeting reports. We developed flexible, semi-structured data collection tools to guide the interviews. The topic guides were developed in consensus with the international SPICES consortium using the CFIR interview guide tool [45], and topics were adapted (Passive voice) to our local context and stakeholder groups. The interview guides developed for this study are provided (Passive voice) in Appendix A. An experienced team of qualitative researchers (NA, SA, HB) collected data until data sufficiency was obtained (Passive voice). At least two researchers were present as moderator moderators or observer  observers in each focus group. All interviews were held face-to-face and were audio-recorded. The interviewers took field notes of their experiences during data collection.

·        This study explored the views of macro, meso and micro level stakeholders on the contextualization of a comprehensive intervention program for the primary prevention of CVD, along with determinants to its implementation in PHC and community settings in a Belgian urban context. This pre-implementation study was carried out as part of H2020 SPICES project, since contextual factors may be important in the implementation of the EBSP. This pre-implementation study was carried out as part of the H2020 SPICES project since contextual factors may be necessary for implementing the EBSP. The CFIR enabled the identification of determinants, barriers and facilitators across its domains and constructs, providing an opportunity to inform further design of intervention components and implementation strategies for implementation in new settings in the next steps of the project. The CFIR identified determinants, barriers and facilitators across its domains and constructs, providing an opportunity to inform further design of intervention components and implementation strategies for implementation in new settings in the project's next steps.

·        Macro-, meso- and micro-level stakeholders’ views demonstrated various contextual dimensions that will need to be considered (Passive voice) when implementing a comprehensive program comprising of complex interventions for the primary prevention of CVD in PHC and community settings and underscored several criteria that seem necessary to transform health systems towards a network-oriented approach of health and wellbeing. These results form a solid foundation to tailor the H2020 SPICES project to the needs and preferences of the target population and potential implementers, but also and to better respond to policy evolutions. The next steps in our research project can clarify how these complex and dynamic determinants are interrelated and how they influence the outcomes and process of actual implementation of implementing the EBSP in real-life settings. Ongoing stakeholder engagement is needed to develop sustainability in this multi-dimensional, multilevel and, dynamic field.
